# Fast and Accurate Multi-Task Learning for Encrypted Network Traffic Classification

Jee-Tae Park [1] , Chang-Yui Shin [2], Ui-Jun Baek [1] and Myung-Sup Kim [1,*]

1 Department of Computer and Information Science, Korea University, Sejong 30019, Republic of Korea; pjj5846@korea.ac.kr (J.-T.P.); pb1069@korea.ac.kr (U.-J.B.)
2 C4ISR System Development Quality Research Team, Defense Agency for Technology and Quality, Daejeon 35409, Republic of Korea; superego99@dtaq.re.kr
* Correspondence: tmskim@korea.ac.kr

**Abstract:** The classification of encrypted traffic plays a crucial role in network management and security. As encrypted network traffic becomes increasingly complicated and challenging to analyze, there is a growing need for more efficient and comprehensive analytical approaches. Our proposed method introduces a novel approach to network traffic classification, utilizing multi-task learning to simultaneously train multiple tasks within a single model. To validate the proposed method, we conducted experiments using the ISCX 2016 VPN/Non-VPN dataset, consisting of three tasks. The proposed method outperformed the majority of existing methods in classification with 99.29%, 97.38%, and 96.89% accuracy in three tasks (i.e., encapsulation, category, and application classification, respectively). The efficiency of the proposed method also demonstrated outstanding performance when compared to methods excluding lightweight models. The proposed approach demonstrates accurate and efficient multi-task classification on encrypted traffic.

**Keywords:** encrypted traffic classification; multi-task classification; BERT; transformer

## 1. Introduction

The advancement of science and technology and ultra-high-speed networks is accompanied by the rise of various applications. With the advancement of modern network technologies such as cloud computing and edge computing, research on efficient network management has been actively conducted. Among them, network traffic classification research is one of the key factors for efficient network management [1–5].

Traffic classification methods encompass traditional, signature-based, learning-based, and transformer-based approaches [3–8]. Traditional methods rely on port-based and payload-based techniques. Port-based classification uses origin and destination ports, offering simplicity and low computational cost, but it faces limitations with dynamic ports. Payload-based classification utilizes fixed payload content, providing simplicity and high performance, but it is susceptible to encrypted traffic and struggles to adapt to new protocols [9]. Signature-based methods classify traffic based on specific patterns or signatures, demonstrating high performance for defined signatures. However, they face challenges in adapting to changing patterns and encrypted traffic. Overall, network traffic classification research plays a key role in enhancing efficient network management amid evolving technological landscapes.

With recent advances in AI and technologies, most studies are using learning-based methods [10–32]. Learning-based methods utilize machine learning (ML) and deep learning (DL) algorithms to learn and classify traffic. Models are trained on large amounts of traffic data to identify specific patterns or trends, which are then used to predict or classify new traffic. Due to these advantages, many studies have utilized learning-based methods, and they have improved performance in many areas.

Transformer-based methods are one of the more recent deep learning techniques to emerge, applying structures that have performed particularly well in natural language processing (NLP) for traffic classification [33–36]. The self-attention mechanism of the transformer effectively learns the global dependencies of sequence data, which has shown promising performance in a variety of applications. For instance, the field of NLP has witnessed a notable advancement with the introduction of bidirectional encoder representation from transformers (BERT) pre-training models [35,36]. BERT has demonstrated high performance in many fields and can be effectively applied to downstream tasks by learning relationships and structures for unbiased data from unlabeled data. In line with this trend, many studies have been conducted in the field of network traffic classification by applying transform-based methods. These methods have shown higher performance than traditional learning-based methods.

With the growing concerns regarding personal privacy and security, most applications now utilize encrypted traffic [37–39]. As encrypted communications protect payload content, traditional traffic classification methods have become inapplicable. Researchers use publicly available encrypted traffic datasets such as ISCX 2016 VPN/Non-VPN [40]. for encrypted traffic classification studies. In these encrypted traffic classification studies, public datasets are mainly divided into intrusion detection systems (IDS) and application classification, each of which is in turn divided into specific tasks. For example, the ISCX 2016 VPN/Non-VPN, which is often used for application classification studies, consists of three tasks: encapsulation, category, and application.

Traffic classification methods are categorized into single-task learning (STL) and multi-task learning (MTL) based on the target data task. STL focuses on training a model for a specific task in machine learning, enhancing performance by learning task-specific features and patterns. However, this optimized model may have limited applicability to other tasks. On the other hand, MTL involves training a model on multiple related tasks, utilizing shared representations to improve overall performance. MTL shares common low-level features across tasks while incorporating task-specific high-level features. This approach is valuable for diverse yet interrelated tasks, leading to more efficient and effective learning [41–43].

Most network traffic classification research has traditionally used STL, and while classification performance has improved, there are some limitations to applying traditional STL. First, the evolving complexity of networks, including intricate network traffic patterns, new network environments, applications, and encryption technologies, has challenged the applicability of traditional STL. Second, STL requires training a separate model for each task, which is time and resource intensive. Third, malicious activity on the network is becoming increasingly sophisticated. Attackers are adept at evading or defeating traditional security methods, requiring more detailed analysis that is more diverse and broader than traditional research. Therefore, it is essential to study traffic classification with MTL, which can address the limitations of traditional research by analyzing network traffic more comprehensively and in-depth compared to STL.

In this paper, we propose a multi-task classification method utilizing DistilBERT [36], a variant of the BERT model within a transformer architecture, for classifying encrypted traffic. This approach enables the performance of traffic classification for various tasks with a single training, using BERT. Our contributions can be summarized as follows:

- We adopt a multi-task learning (MTL) approach for encrypted traffic classification, leveraging the DistilBERT model. The proposed method is based on a model that can handle multiple classification tasks simultaneously. The proposed method allows for a thorough and detailed analysis of encrypted network traffic, addressing the complexity of various tasks within a unified training framework.
- To validate our proposed method, we conducted verification experiments, focusing on three specific tasks using the ISCX 2016 VPN/Non-VPN dataset. We compared our approach with other methods, assessing classification accuracy and efficiency. In terms of classification accuracy, we demonstrated average accuracies ranging from

96.89~99.29% across all tasks, outperforming the majority of existing methods. In terms of model efficiency, our approach showed favorable per sample processing time compared to existing models. Through our experiment results, we validate that our proposed method, employing multi-task classification for encrypted traffic, is effective in terms of both classification performance and efficiency.

- We applied weight adjustments (class weight, task weight) within the model to solve the problems related to data imbalance and varying task difficulty. Through additional experiments, we validated the impact of both weights on performance improvement. This underscores the effectiveness of our approach in diverse scenarios, enhancing its applicability across various situations.

The remainder of this paper is organized as follows. In Section 2, we will describe the related work, and in Section 3, we will provide a detailed explanation of the proposed method. In Section 4, we conduct an experiment by using the ISCX 2016 VPN/Non-VPN dataset, including a multi-task classification experiment, and we will discuss several issues in Section 5. Finally, we conclude the paper and outline future research directions in Section 6.

## 2. Related Works

### 2.1. Overview of the Network Traffic Classification

Network traffic classification research is the study of analyzing the traffic generated by computer communications, which is essential for the effective management, monitoring, and security of computer networks. As shown in Figure 1, network traffic classification research is broadly classified according to the field of research, methodology, classification level, and data units processed.

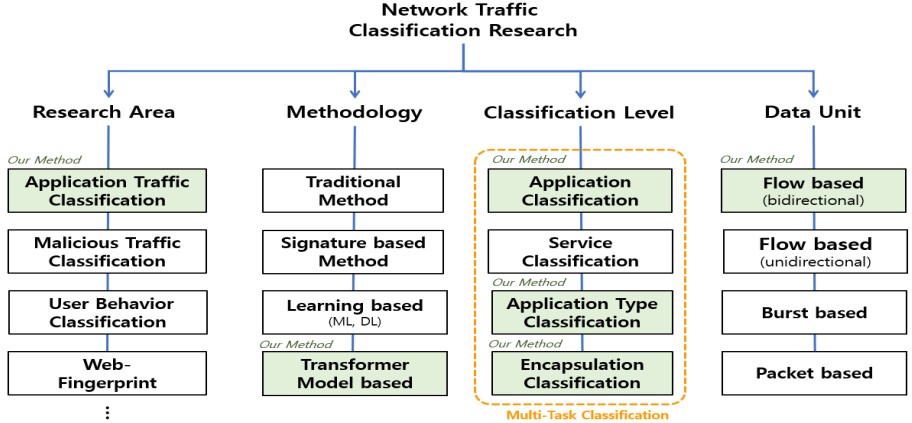

**Figure 1.** Overview of the network traffic classification.

First, in terms of research areas, it consists of various subfields, including application classification [10–26], malicious traffic detection [29–32], user behavior profiling [27–30], and web fingerprinting [44–46], of which application classification and malicious traffic detection are the most widely studied. Second, in terms of methodologies, methods such as port-based and payload-based methods have traditionally been widely used. Port-based classification categorizes traffic based on known port numbers, which is inapplicable because many applications use dynamic ports. Payload-based methods classify applications based on fixed payload content. Signature-based methods extend the mechanisms of payload-based methods to various traffic characteristics, defining common statistical, header, and behavioral characteristics of traffic as signatures and classifying based on them. Both payload-based and signature-based methods perform poorly on encrypted traffic. To solve these limitations, learning-based methods using machine learning and deep learning are the most active, and recently, methods using transformer models have also been performed. Third, in terms of classification level, it consists of the following levels: application classification, which distinguishes each application; service classification, which

categorizes the detailed features, services, and behaviors of the application; application type classification, which categorizes the characteristics of the application such as Chat or File Transfer; and encryption classification, which categorizes the presence or absence of encryption. Fourth, in terms of data units, it is categorized into unidirectional and bidirectional flows, packets, and bursts. A flow is a set of packets with the same 5-tuples of information in the packet header, and a burst is a set of time-adjacent network packets originating from either the request or the response in a single-session flow [34].

As mentioned before, we propose a multi-task classification method for encrypted traffic using DistilBERT to perform encapsulation, application type, and application classification on ISCX 2016 VPN/Non-VPN data. In Figure 1, the green-colored parts represent the four aspects of our proposed method.

### 2.2. Encrypted Traffic Classification

Network traffic classification has been around for a long time and has primarily utilized traditional methods based on port and payload, as well as signature-based methods. However, traditional traffic analysis methods are ineffective because many modern applications, including mobile, cloud, and IoT, rely primarily on encrypted traffic. To address the limitations of traditional methods, recent research has turned to learning-based approaches involving ML and DL [10–25].

In [10], Lotfollahi et al. introduced Deep Packet, a system that utilizes a stacked autoencoder and CNN. They achieved an impressive F1 score of 98% for application identification on the ISCX 2016 VPN/Non-VPN dataset. Wang et al. [11] introduced a novel method to convert packets into images and process them using 1D-CNN, which showed promising results on ISCX 2016 VPN/Non-VPN. In [12], Zou et al. pre-sent an encrypted network traffic classification approach using CNNs and LSTM networks; in [13], they proposed an innovative fusion of CNNs and designed RNNs for service recognition in IoT traffic; in [14], they used naïve Bayes, C4. 5 decision trees, Bayesian networks, and naive Bayes trees. They performed a comprehensive analysis comparing the performance of these algorithms using 22 features extracted from network flows. In [15], they introduced flow sequence network (FS-NET) for encrypted traffic classification. FS-NET utilizes both RNNs and a multi-layer encoder–decoder structure. In [16], the authors proposed FlowPic, a classification method that converts consecutive packet sizes in a flow into a two-dimensional gray image and uses CNNs for classification. While FlowPic is simple and performs well, it is not suitable for real-time traffic classification because it requires the capture of traffic over a long period of time. The authors also note that it is not applicable to classifying some encrypted traffic. In [17], the authors proposed TSCRNN, which automatically extracts features for efficient traffic classification based on spatiotemporal features. To validate the proposed method, the authors conducted experiments on ISCX Tor 2016 data and obtained high accuracy. In [18], the authors proposed MIMETIC, which exploits traffic data heterogeneity by learning both intra- and inter-modality dependencies to overcome performance limitations. MIMETIC outperforms single-modality DL-based, state-of-the-art ML-based mobile traffic classifiers. In [19], the authors propose an improved DAGSVM classification method by focusing on the error accumulation of the traditional DAGSVM algorithm. Experimental results show that the proposed method has higher classification accuracy than traditional DAGSVM while having an acceptable time cost. The studies in [39] and [47] have conducted research with a focus on lightweight models rather than classification performance. While most studies primarily emphasize performance, they highlight the importance of lightweight approaches for handling large-scale traffic data.

In recent years, there has been a surge in research centered on transformer architectures characterized by self-attention and multi-headed attention mechanisms. Transformer-structured models mainly utilize the BERT model, which has proven to show strong performance in the NLP field, but recently, research has also been conducted using the masked autoencoder (MAE), which is used in the CV field [33–35].

In [34], the authors proposed ET-BERT, a novel approach inspired by transformer architectures. It presents a new pre-training method designed for encrypted traffic classification and fine-tuned for optimal performance achieving an accuracy of over 97%. In [21], the authors propose a method called PERT (payload encoding representation from transformer) utilizing dynamic word embedding. PERT outperforms other methodologies on publicly available encrypted traffic datasets and captures Android HTTPS traffic. In [22], the authors propose the BFCN model, which combines BERT and CNN models to derive global traffic features with a pre-trained BERT model and byte-level local traffic features with a CNN model. The experimental results show F1 scores of 99.11% and 99.41% in the traffic service and application identification tasks operating on the ISCX 2016 VPN/Non-VPN dataset, respectively. In [23], similar to [22], a pre-trained BERT model and a bidirectional LSTM are applied together, with an accuracy of about 99%. In [33], the authors utilize DistilBERT to perform encrypted traffic classification research. They introduce comparative learning to enhance classification speed without degrading performance. Although our study is similar to [33], which focuses on STL, our study specifically targets MTL. We apply MTL to simultaneously learn three tasks on a single model, resulting in superior performance.

In [24,25], both studies utilize MAE for traffic classification research. The authors propose a pre-training model for MAE that introduces a mask patch model, a self-supervised learning pre-training task, to capture unbiased representations from bursts of varying lengths and patterns. Experiment results show that the proposed system achieves new high levels of accuracy of 98%, classification speed, memory efficiency, and robustness across a wide range of network traffic types.

## 2.3. Overview of the Multi-Task Learning

The advent of deep learning has led to significant performance improvements in CV and NLP, as well as network traffic classification. The typical approach is to learn these tasks in isolation, where a separate neural network is trained for each individual task [15–25]. Nevertheless, deep learning-based methods suffer from a number of limitations in terms of time and memory. Recently, research has been conducted on MTL techniques, which have shown promising results in terms of performance, computational, and/or memory efficiency [41–43]. MTL is the joint handling of multiple tasks through a learned shared representation. In [41], the author introduces hard parameter sharing and soft parameter sharing and discusses techniques such as deep relationship networks and fully adaptive feature sharing. In [42], the authors investigate various aspects of MTL. First, we provide a definition of MTL, and then we categorize supervised MTL models into five main approaches and discuss their characteristics The authors note that outlier tasks that are unrelated to other tasks are known to degrade the performance of all tasks when learning collaboratively, and they present this as a challenge. In [43], the authors present an overview of architectural and optimization-based strategies for MTL within the scope of deep neural networks. They also introduce how to set weights for each task in an MTL. In summary, MTL leverages useful information from multiple related tasks with the goal of improving the generalization performance of any task. MTL is efficient in terms of performance, time, and memory as it can handle multiple tasks using a single model. However, it is important to consider the correlation between tasks, the structure of the model, and optimization because certain tasks can degrade the performance of others.

With the rising interest in MTL, there is a gradual increase in research applying MTL to traffic classification studies [48–50]. In [48], the authors claim to be the first to apply MTL in network traffic classification research and utilize CNNs to perform malware detection. In [49], the authors employ three time-series features and utilize CNN for multi-task classification on QUIC and ISCX 2016 VPN/Non-VPN datasets. However, the detection performance appears relatively low with an accuracy range of 82–92%. The classification task is configured slightly differently compared to previous studies. In [50], the authors perform multi-task classification using transformer and 1D-CNN, achieving an accuracy of 97–98% on the ISCX 2016 VPN/Non-VPN dataset. Our work is similar to their work.

Their study is similar to ours, but we demonstrate accuracy exceeding 99% across all three tasks. Additionally, we evaluate the efficiency of multi-task classification, an aspect not addressed in their work.

## 3. Proposed Method

### 3.1. Model Architecture

The entire system structure consists of three sub-systems (i.e., data preprocessing, byte tokenizing, and multi-task classification) and is shown in Figure 2. Data preprocessing is the process of converting raw traffic data into an input format before applying it to DistilBERT model, resulting in byte-separated data as the output. Byte tokenizing takes the data from the previous module as the input and performs tokenization for each byte. Multi-task classification takes the tokenized data as the input, performs embedding, runs it through the DistilBERT model, and predicts a label for each task.

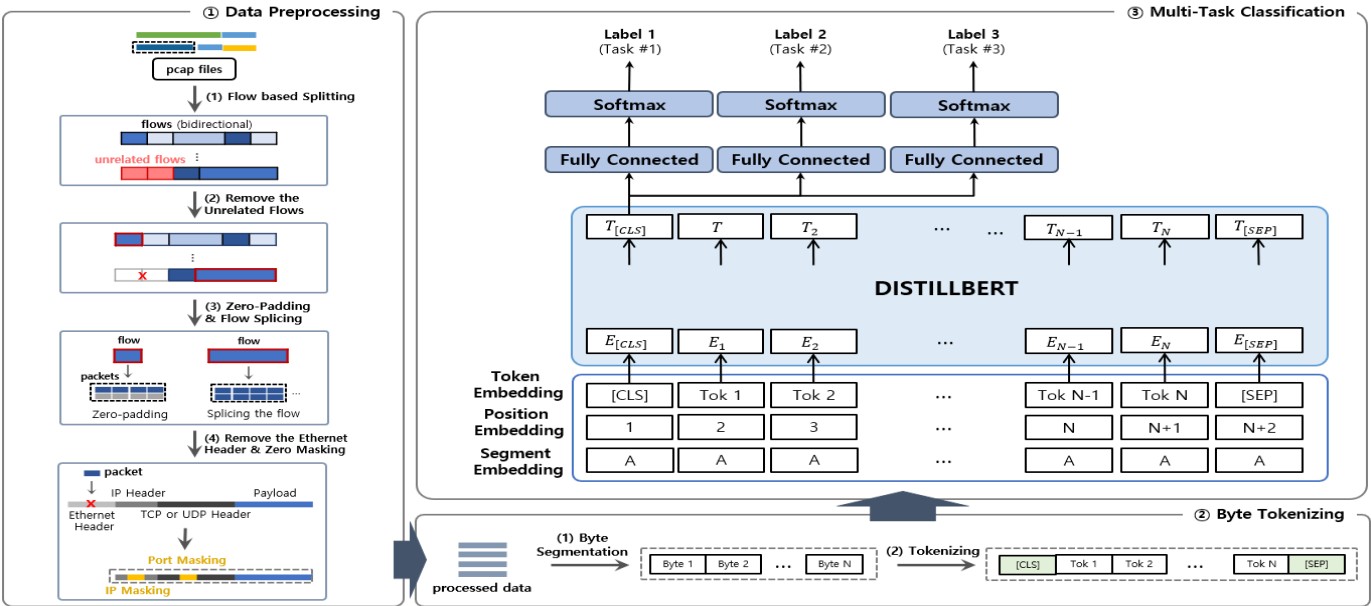

**Figure 2.** Architecture of the proposed method.

### 3.1.1. Data Preprocessing

(1) Target Dataset: While there have been many publicly available network traffic datasets for a long time, encrypted traffic datasets are the most common. There are several encrypted traffic datasets available, but we use the ISCX 2016 VPN/Non-VPN dataset [40], which is the most popular in this research area. This dataset is captured from real traffic and is a publicly available dataset in raw pcap format consisting of traffic from various applications. Since it is the most popular dataset used in several previous studies, it allows for the comparison and interpretation of experimental results from multiple studies. The dataset is broadly categorized into three classes (i.e., encapsulation, category, and application), and separate classification studies are typically performed for each label. Table 1 shows information about the classes for each task. Encapsulation refers to the presence or absence of encryption on the target traffic and consists of two classes: VPN and Non-VPN. Category refers to the nature of the application and consists of six classes, excluding web browsing. Application indicates the application used and consists of sixteen classes.

**Table 1.** Class information for three tasks in ISCX 2016 VPN/Non-VPN dataset.

| Task | Classes |
|---|---|
| Encapsulation (2) | VPN, Non-VPN |
| Category (6) | Chat, Email, Streaming, File Transfer, P2P, VoIP |
| Application (16) | Skype, ICQ, Hangout, Facebook, Email, Gmail, FTP, SFTP, SCP, Netflix, Spotify, Vimeo, YouTube, AIM Chat, VOIPBuster, BitTorrent |

(2) Preprocessing: We perform the following preprocessing. First, we convert the packet-level pcap file to flow-level. We segment the capture files into bidirectional flows using the SplitCap tool. Second, we remove irrelevant flows from the converted flow file. The ISCX 2016 VPN/Non-VPN dataset contains approximately 309 K flows in total. However, as noted in [51], the dataset contains a lot of irrelevant flows. For example, it also includes traffic that is not application-specific, such as NBSS, LLMNR, DNS, etc. and the disrupted three-way handshake flows. Through the preprocessing steps outlined in [51], a total of 29,195 flows were identified. We performed further analysis and found that there were specific flows within these flows, characterized by UDP, a destination IP of 255.255.255.255, and a consistent inclusion of the string "Beacon~" in the payload. These flows were considered non-essential for the research objectives; therefore, we removed these unnecessary flows from the converted flow data. After going through the first and second process, we finally obtained 8763 flows. Third, we performed zero-padding and flow splicing from the converted data. Considering the subsequent byte tokenization process, we extract 63 bytes from each of the eight packets in the flow. In this process, if the number of bytes in a packet is less than 63, we perform zero-padding. If the packet has more than 63 bytes, we perform splicing. Based on other research [33,34] and experiments under various configurations, we chose 63 as the optimal byte value. The 63 bytes are composed of (1) IP, (2) TCP or UDP, and (3) Payload, depending on the network layer and data. In this case, the IP has the same number of bytes at 20 bytes, but the lengths of the headers for TCP and UDP are 20 and 8 bytes, respectively, so the length of the payload that comes after it will be different. Therefore, the UDP header is extended to 20 bytes by using zero-padding at the end. We also perform zero-padding for flows that are less than 63 bytes in length for the entire flow, and in the case of UDP, additional padding is performed for the UDP header. Finally, we remove the Ethernet header and, masking the IP, port to zero. These are masked as it can cause biased interpolation as it has strong identifying information. Figure 3 shows the distribution of bidirectional flows by class for pre-processed data. In Figure 3, we can see that the three tasks suffer from data imbalance between each class, which we address in Section 3.2.1.

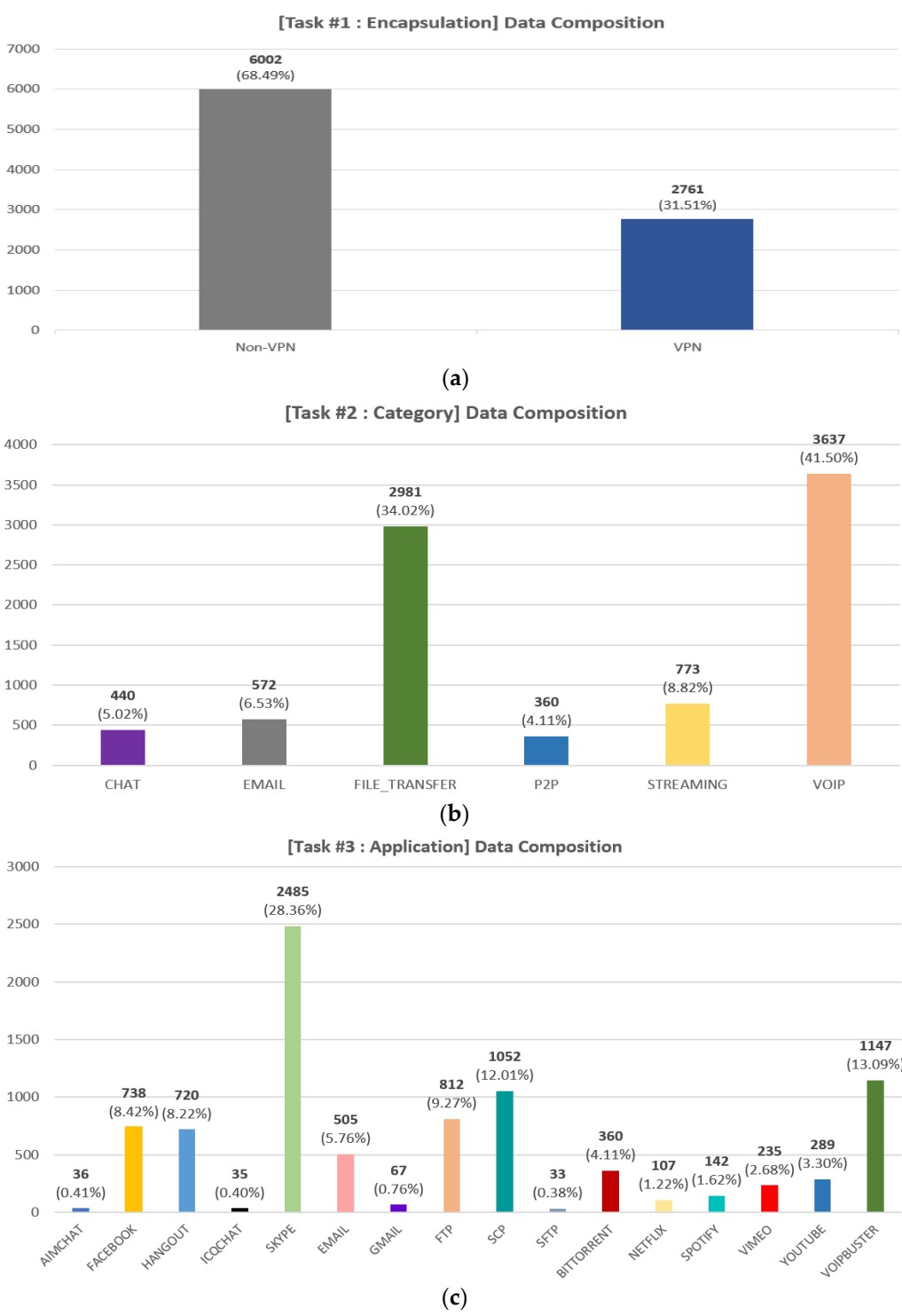

**Figure 3.** Data composition of the pre-processed data for the three tasks: (**a**) encapsulation task (two classes), (**b**) category task (six classes), (**c**) application task (sixteen classes).

### 3.1.2. Byte Tokenizing

Byte tokenizing is the process of separating preprocessed data into bytes and converting the separated bytes into tokens. There are two parts to this process: First, we split the preprocessed data into bytes to use as the input. Second, the process of converting the extracted bytes of data into tokens is performed. In this process, it is crucial to determine the number of tokens to be used for organizing the data. If the number of tokens is too high, it may increase the data processing load, while too few tokens can result in the loss of

essential information for classification, leading to performance degradation. Additionally, considering that BERT can handle a maximum of 512 tokens, selecting an appropriate number of tokens is essential. After experimenting with various combinations, we ultimately chose 63 bytes for the first eight packets, which can accommodate a total of 506 tokens, including two special tokens [CLS] and [SEP]. We present a performance comparison based on input shape in Section 5.3.

### 3.1.3. Multi-Task Classification

BERT is an NLP model that utilizes a transformer-based architecture and excels in bidirectionally understanding context within sentences. It encompasses two phases: pre-training and fine-tuning. In the pre-training stage, BERT undergoes immersion in extensive amounts of unlabeled data. This process involves two phases: next sentence prediction (NSP) and masked language modeling (MLM). In the NSP phase, the model learns to predict whether a sentence follows another sentence in the input text, enhancing its grasp of discourse-level context. In the MLM phase, certain words in the input sentences are randomly masked, and model is trained to predict these masked words, fostering a bidirectional understanding of context at the word level. In the fine-tuning phase, the pre-trained BERT model is further refined for specific tasks, such as text classification or question answering, optimizing the process for each task. In network traffic classification research, a large amount of unlabeled traffic is collected in a pre-training phase to learn the structure and relationships within the traffic. Each downstream classification task is then performed in a fine-tuning phase. In [33], pre-training was performed using about 30 GB of unlabeled traffic data, and five tests were performed with fine-tuning.

Our proposed method does not utilize an additional pre-training model and directly uses the fine-tuning model of DistilBERT. This is because in the field of network traffic classification, the pre-training process has several limitations. First, the traffic structure is very diverse and extensive, but the input dimensions of the BERT model are limited. Second, the temporal and spatial features in the packet header are ignored, resulting in performance degradation. These limitations make it difficult for the model to fully learn the characteristics of different network traffic. Third, the pre-training process is computationally intensive, requiring substantial time, memory overhead, and high-performance hardware due to the utilization of extensive traffic data. In addition, we perform byte-level tokenizing as in [51]. As the authors of [51] note, the values derived from the previous traffic preprocessing and byte tokenizing are represented as integers between 0 and 255, allowing us to directly fine tune the DistilBERT [36] model, which is explicitly provided as "distilbert-base-uncased".

The output layer uses [CLS] as the final sequence representation for downstream task classification. The [CLS] token output may be converted into a class probability based on the task. MTL predicts multiple task labels from [CLS] tokens, with approaches such as hard parameter sharing (tasks share all parameters) and soft parameter sharing (tasks have their own parameters, sharing some). Hard parameter sharing is efficient with shared parameters, suitable for related tasks, while soft parameter sharing allows task specialization for tasks with diverse characteristics.

Therefore, it is important to consider the relevance and nature of the task within the target dataset and choose the appropriate method. As mentioned before, we target three different tasks in the ISCX 2016 VPN/Non-VPN dataset, and all three tasks are related to each other as they perform task-specific classification on the same data. Therefore, we utilized the hard parameter sharing for MTL, and Figure 4 shows the proposed MTL structure.

Figure 4 is organized into shared layers and task specific layers, where the model and different parameter sets are shared in the shared layer, and the task-specific layers are used to classify and derive results for each task. The shared layers include the embedding layer and the transformer encoding layer used by the DistilBERT model.

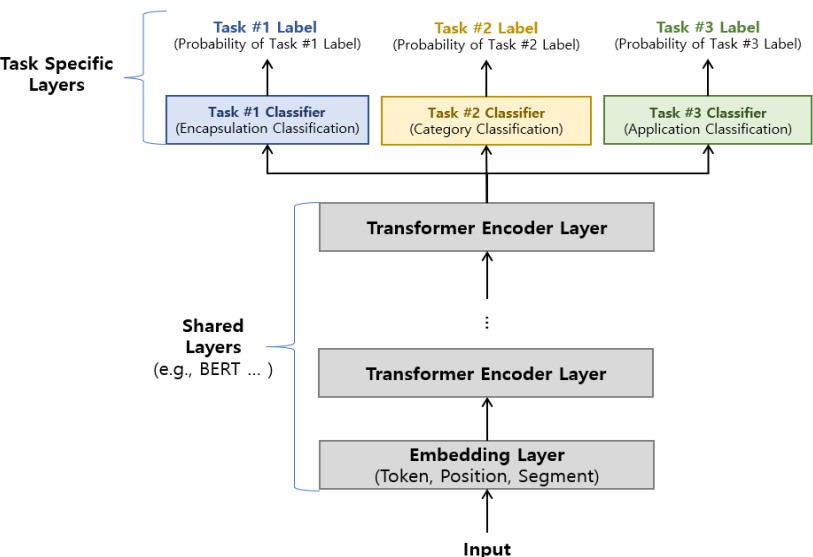

**Figure 4.** Structure of the proposed MTL.

*3.2. Weight Adjustment*

3.2.1. Class Weight for Imbalanced Data

As shown in Figure 3, the data are heavily imbalanced. Data imbalance stands as a significant challenge constraining the performance of ML models, particularly when the samples of the minority class are insufficient [52,53]. To address this issue, common practices involve the utilization of undersampling and oversampling techniques. However, these methods come with risks of underfitting and overfitting, respectively, potentially limiting the generalization ability of the model.

$$W_{kj} = 1 - \frac{C_{kj}}{\sum_j C_{kj}} \tag{1}$$

In recent research, weighted classes have been recognized as one approach to addressing data imbalance [33]. Weighted classes can significantly reduce the bias in the data; thus, we utilize a method for calculating class weights. Equation (1) indicates the method for calculating the normalized weights for each class. In Equation (1), $W_{ki}$ is the weight for each class in task $k$, $C_{ki}$ is the number of samples for each class label within the $k$ tasks, $k$ indicates target task, and $j$ indicates class label. These weights are utilized to adjust the training of the model, taking into consideration the imbalance within each class, thereby aiding in enhancing the overall model performance.

3.2.2. Task Weight for Loss Calculation

In a typical DL, loss is a metric that represents the difference between the model's predictions and the actual target. Minimizing this difference allows the model to learn the desired outcome more effectively. Loss is often calculated through an objective function (loss function), most commonly the cross-entropy, mean squared error, etc. In multi-task classification, the loss is different for each task, so it is necessary to calculate the loss for each task step by step and combine them effectively to obtain the final loss. Equations (2) and (3) indicate the method for accumulating losses in multi-task classification. In Equation (2), $y\prime_i$ is the model's predicted value, $y_i$ is the actual value, and $f_i$ is the objective function for task $i$. After calculating the loss for each task, they are combined to obtain the final loss. In Equation (3), *Total Loss* is the final loss, which is the aggregate of the losses from each task, $N$ is the number of tasks, and $\alpha_i$ is a weight that represents the relative importance of each task.

$$L_i = W_{kj} \times f_i(y\prime_i, y_i) \tag{2}$$

$$Total\ Loss = \sum_{i=1}^{N} \alpha_i \times L_i \tag{3}$$

In MTL, performance and learning time can vary due to differences in the difficulty of each task. Typically, easier tasks converge quickly to achieve high accuracy, while more difficult tasks face complications in convergence and require more extensive training. Allocating equal weights to all tasks in MTL may not be appropriate, as it could lead to higher weights for easier tasks, diminishing the model's learning capacity for difficult tasks. Therefore, in MTL, it is essential to consider the difficulty of each task and assign appropriate weights. Equation (4) illustrates a method for determining the weights for each task in light of their respective difficulties.

$$\alpha_i = \frac{E_i}{\sum_{i=1}^{N} E_i} \tag{4}$$

In Equation (4), $E_i$ represents the minimum number of epochs required to converge to performance β. β is measured by accuracy and can be dynamically adjusted. However, continuous weight adjustments may decrease the model's stability and increase the risk of overfitting to specific tasks. Therefore, we set β to 90% through various experiments. For example, assuming that there are four tasks and it takes 5 epochs in task #1, 10 epochs in task #2, 15 epochs in task #3, and 20 epochs in task #4 to achieve 90% accuracy each, the weights are set to 0.1 (5/50), 0.2 (10/50), 0.3 (15/50), and 0.4 (20/50), respectively.

## 4. Evaluation, Result and Analysis

### 4.1. Evaluation Environment Setup

The proposed method was implemented using Python 3.10.9 and PyTorch 2.0.1 with CUDA 11.8. All experiments were performed on a Linux Ubuntu 20.04.6 LTS server with a 24-core Intel(R) Core(TM) i9-10920X CPU (3.50 GHz) and NVIDIA GeForce RTX 4090 GPU (24 GB memory). We set the optimal parameters for the model through various experiments. We set the learning rate to $2 \times 10^{-5}$, the batch size to 16, and the dropout ratio to 0.1 and used AdamW as the optimization tool. Each dataset is divided into the training set and the testing set according to the ratio of 7:3. We randomly selected 500 samples from each task (6 categories, 16 applications in total) and entered them into the dataset; however, if the number of samples for some applications (e.g., Gmail, SFTP within an application classification) was less than 500, we selected all samples for that application.

### 4.2. Evaluation Metrics

When evaluating the performance of a model, the evaluation metrics are important. We utilized four evaluation metrics that have been used in several studies: accuracy, recall, precision, and F1 score. Equations (5)–(8) show the method for calculating these metrics

$$Accuracy = \frac{TP + TN}{(TP + FN + FP + TN)} \tag{5}$$

$$Recall = \frac{TP}{(TP + FN)} \tag{6}$$

$$Precision = \frac{TP}{(TP + FP)} \tag{7}$$

$$F1\ Score = \frac{2 \times Recall \times Precision}{(Recall + Precision)} \tag{8}$$

True positive (TP) is when the model correctly classifies something as positive, and true negative (TN) is when the model correctly classifies something as negative. False positive (FP) is when the model incorrectly classifies something as positive when it was negative, and false negative (FN) is when the model incorrectly classifies something as negative when it was positive.

As previously mentioned, the ISCX 2016 VPN/Non-VPN data are highly imbalanced between classes. To account for the potential bias in the results due to the imbalance between the different categories of data, we used macro average [36]. Macro average calculates the average value of precision, recall, accuracy, and F1 scores for each category to provide a more comprehensive and unbiased assessment across all categories.

### 4.3. Evaluation Result

In this section, we describe our experiments and results to validate the proposed method. We present the classification performance of our proposed model in Section 4.3.1 and conduct a performance comparison with other models in Section 4.3.2. We validate the efficiency of our proposed method in Section 4.3.3 and describe several discussions in Section 5.

#### 4.3.1. Performance of the Proposed Method

To validate our proposed method, we performed experiments on three tasks. In task #1, classifying the encryption, the highest accuracy, precision, recall, and F1 score were 99.29%, 98.61%, 99.47%, and 99.03%, respectively. In task #2, classifying the category, the highest accuracy, precision, recall, and F1 score were 97.38%, 97.31%, 95.93%, and 96.61%, respectively. In task #3, classifying the application, the highest accuracy, precision, recall, and F1 score were 96.89%, 96.91%, 95.13%, and 96.01%, respectively.

Figure 5 illustrates the confusion matrix detailing accuracy for each task. In subfigures (a), (b), and (c), the confusion matrix is presented for each task. In Figure 5, while the majority of classes within each task demonstrate a high accuracy exceeding 95%, AimChat and ICQChat in Figure 5c exhibit relatively lower accuracy. These applications, designed for online chatting and offering various services like voice and video calls, share common traits. However, the similarities between these applications make it difficult to distinguish traffic patterns accurately, leading to decreased classification accuracy. The intricacies of these chat applications contribute to the difficulty in achieving higher performance.

Table 2 shows the best class segmentation results for evaluation performance by class within each task, and 50 epochs in total were performed for the experiment. Task #1 involves classifying two classes (i.e., VPN and Non-VPN), resulting in 98~99% accuracy, precision, recall, and F1 score. Task #2 involves categorizing traffic into six classes (i.e., Chat, Email, File Transfer, P2P, Streaming, VoIP). In task #2, the classes Email, P2P, Streaming, and VoIP are classified with 98~100% accuracy, while File Transfer and Chat are classified with relatively low accuracy of 95.44% and 94.86%. Task #3 involves categorizing traffic into sixteen classes (i.e., AimChat, Facebook, Hangout, ICQChat, Skype, Email, Gmail, FTP, SCP, SFTP, BitTorrent, Netflix, Spotify, Vimeo YouTube, VoIPBuster). In Task #3, most classes were classified with 96–100% accuracy, with some relatively low accuracy results for certain classes such as Aim Chat, ICQ Chat, FTP, and SFTP.

**Table 2.** Performance for three tasks of ISCX 2016 VPN/Non-VPN Classification.

| Proposed Method | | | | | |
|---|---|---|---|---|---|
| **Task** | **Class** | **Accuracy (%)** | **Precision (%)** | **Recall (%)** | **F1-Score (%)** |
| Task #1:<br>Encapsulation | VPN | 99.45 | 98.72 | 99.87 | 99.29 |
| | Non-VPN | 98.69 | 99.72 | 97.23 | 98.46 |
| Task #2:<br>Category | Chat | 94.86 | 97.65 | 94.86 | 96.21 |
| | Email | 98.21 | 96.90 | 98.21 | 97.55 |
| | File Transfer | 95.44 | 97.58 | 95.44 | 96.50 |
| | P2P | 100.00 | 100.00 | 100.00 | 100.00 |
| | Streaming | 99.41 | 99.71 | 99.41 | 99.56 |
| | VoIP | 97.99 | 96.06 | 97.99 | 97.02 |

**Table 2.** *Cont.*

| | | Proposed Method | | | |
|---|---|---|---|---|---|
| Task | Class | Accuracy (%) | Precision (%) | Recall (%) | F1-Score (%) |
| Task #3: Application | AimChat | 93.75 | 78.95 | 93.75 | 85.71 |
| | Facebook | 97.62 | 98.97 | 97.62 | 98.29 |
| | Hangout | 98.91 | 98.19 | 98.91 | 98.55 |
| | ICQChat | 72.73 | 88.89 | 72.73 | 80.00 |
| | Skype | 99.61 | 99.12 | 99.61 | 99.36 |
| | Email | 97.96 | 98.97 | 97.96 | 98.46 |
| | Gmail | 96.30 | 89.66 | 96.30 | 92.86 |
| | FTP | 87.78 | 91.30 | 87.78 | 89.51 |
| | SCP | 99.76 | 100.00 | 99.76 | 99.88 |
| | SFTP | 92.31 | 100.00 | 92.31 | 96.00 |
| | BitTorrent | 100.00 | 100.00 | 100.00 | 100.00 |
| | Netflix | 97.67 | 100.00 | 97.68 | 98.82 |
| | Spotify | 95.16 | 95.16 | 95.16 | 95.16 |
| | Vimeo | 99.06 | 98.13 | 99.06 | 98.59 |
| | YouTube | 96.15 | 96.90 | 96.15 | 96.53 |
| | VoIPBuster | 94.01 | 96.90 | 96.15 | 92.88 |

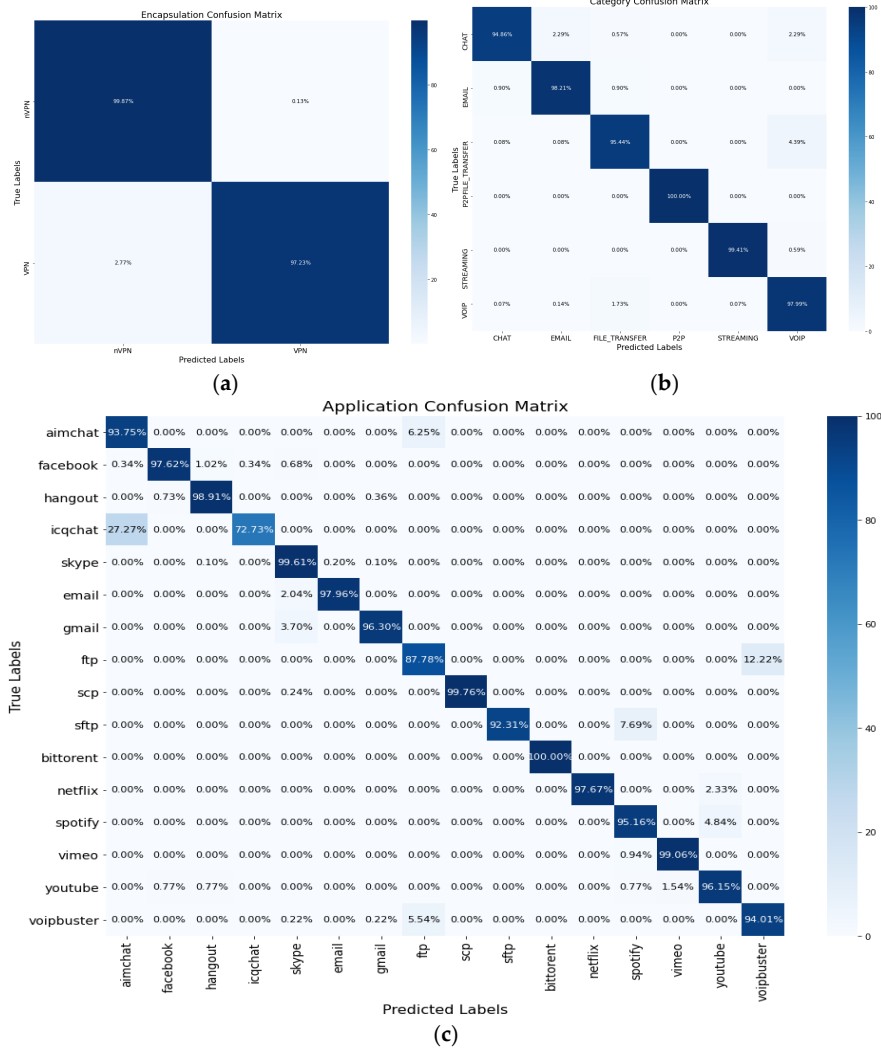

**Figure 5.** Confusion matrix for three tasks: (**a**) encapsulation task (two classes), (**b**) category task (six classes), (**c**) application task (sixteen classes).

Figure 6 shows the learning curve for the three tasks in training and testing. In Figure 6, the losses represent the total losses for the three tasks, with the learning and testing losses gradually decreasing.

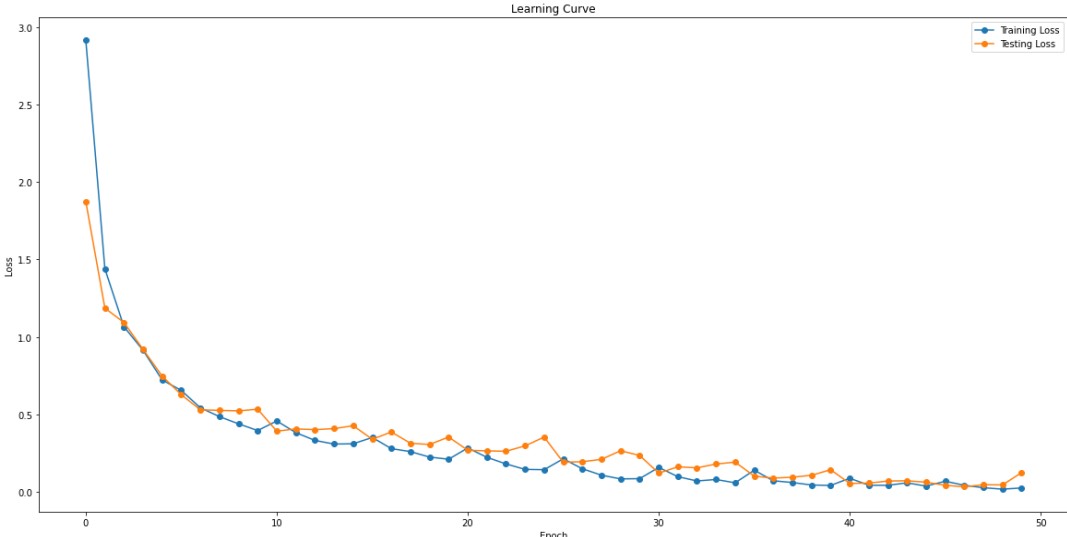

**Figure 6.** Learning curve for the training and testing.

4.3.2. Comparison with Other Model

To validate the performance of our proposed method, we compare its performance with various state-of-the-art methods in network-encrypted traffic classification. For accurate performance validation, it is essential to compare methodologies using the same dataset with identical preprocessing methods in a consistent environment. However, direct comparisons of different methodologies are often impractical due to various constraints. Therefore, we took the performance presented by each methodology and used them for the comparison. The methods are categorized into the following: (1) statistical feature-based, (2) ML- and DL-based, and (3) pretraining-based, and a total of 17 methodologies are compared.

(1) Statistical feature-based methodologies: AppScanner [54], CUMUL [44], BIND [45]
(2) ML- and DL-based methodologies: Deep Fingerprinting (DF) [46], FS-Net [15], Graph-DApp [38], TSCRNN [17], DeepPacket [10], 1D-CNN [26], FastTraffic [39], MATEC [47]
(3) Pretraining-based methodologies: PERT [21], ET-BERT (flow) [34], ET-BERT (packet) [34], XENTC [33], BFCN [22], Flow-MAE [25], YaTC [24]

Most studies do not perform task #1 (Encapsulation) on the ISCX 2016 VPN/Non-VPN data; rather, they perform tasks #2 (Category) and #3 (Application) to classify categories and applications. Therefore, we compare these methods and approaches that target task #2 and #3, as these tasks are more commonly addressed. The results of our experiments are shown in Tables 3 and 4. As each method varies in terms of metrics, number of classes, and targeted tasks, we only summarize the information presented by each study.

**Table 3.** Comparison results for task #2 in ISCX 2016 VPN/Non-VPN.

| Comparison Results for Task #2: Category | | | |
|---|---|---|---|
| **Method** | **Accuracy (%)** | **Precision (%)** | **Recall (%)** | **F1 Score (%)** |
| AppScanner [54] | 71.82 | 73.39 | 72.25 | 71.97 |
| CUMUL [44] | 56.10 | 58.83 | 56.76 | 56.68 |
| BIND [45] | 75.34 | 75.83 | 74.88 | 74.20 |
| DF [46] | 71.54 | 71.92 | 71.04 | 71.02 |
| FS-Net [15] | 72.05 | 75.02 | 72.38 | 71.31 |

| Comparison Results for Task #2: Category | | | |
|---|---|---|---|
| **Method** | **Accuracy (%)** | **Precision (%)** | **Recall (%)** | **F1 Score (%)** |
| GraphDApp [38] | 59.77 | 60.45 | 62.20 | 60.36 |
| TSCRNN [17] | - | 92.70 | 92.60 | 92.60 |
| Deep Packet [10] | 93.29 | 93.77 | 93.06 | 93.21 |
| 1D-CNN [26] | 98.30 | - | - | 98.60 |
| FastTraffic [39] | 94.50 | 94.77 | 94.26 | 94.40 |
| MATEC [47] | 73.20 | 84.43 | 82.40 | 82.87 |
| PERT [21] | 93.52 | 94.00 | 93.49 | 93.68 |
| ET-BERT (flow) [34] | 97.29 | 97.56 | 97.31 | 97.33 |
| ET-BERT (packet) [34] | 98.90 | 98.91 | 98.90 | 98.90 |
| XENTC [33] | 97.03 | - | - | 97.06 |
| BFCN [22] | 99.12 | 99.13 | 99.11 | 99.11 |
| YaTC [24] | 98.07 | - | - | 98.04 |
| Flow-MAE [25] | 99.15 | 99.24 | 99.15 | 99.17 |
| Proposed | 97.38 | 97.31 | 95.93 | 96.61 |

**Table 4.** Comparison results for task #3 in ISCX 2016 VPN/Non-VPN.

| Comparison Results for Task #3: Application | | | |
|---|---|---|---|
| **Method** | **Accuracy (%)** | **Precision (%)** | **Recall (%)** | **F1 Score (%)** |
| AppScanner [54] | 62.66 | 48.64 | 51.98 | 49.35 |
| CUMUL [44] | 53.65 | 41.29 | 45.35 | 42.36 |
| BIND [45] | 67.67 | 51.52 | 51.53 | 49.65 |
| DF [46] | 61.16 | 66.97 | 66.51 | 65.31 |
| FS-Net [15] | 66.47 | 48.19 | 48.48 | 47.37 |
| GraphDApp [38] | 62.28 | 59.00 | 54.72 | 55.58 |
| TSCRNN [17] | - | - | - | - |
| Deep Packet [10] | 97.58 | 97.85 | 97.45 | 97.65 |
| 1D-CNN [26] | 86.60 | - | - | 86.50 |
| FastTraffic [39] | 92.24 | 93.58 | 92.84 | 93.12 |
| MATEC [47] | 69.21 | 73.32 | 65.40 | 68.24 |
| PERT [21] | 82.29 | 70.92 | 71.73 | 69.92 |
| ET-BERT (flow) [34] | 85.19 | 75.08 | 72.94 | 73.06 |
| ET-BERT (packet) [34] | 99.62 | 99.36 | 99.38 | 99.37 |
| XENTC [33] | 96.37 | - | - | 94.63 |
| BFCN [22] | 99.65 | 99.36 | 99.47 | 99.41 |
| YaTC [24] | - | - | - | - |
| Flow-MAE [25] | 99.87 | 99.91 | 99.89 | 99.90 |
| Proposed | 96.89 | 96.91 | 95.13 | 96.01 |

The proposed method achieves about 96~98% accuracy on tasks #2 and #3, outperforming most of the existing research methods. Although several methodologies exhibit slightly better performance (i.e., accuracy 0.69–1.67% in task #2 and accuracy 1.31–2.98% in task #3), it is noteworthy that the existing approaches are designed for STL-based single task classification, while the proposed method is capable of classifying three tasks simultaneously. This capability to address multiple tasks simultaneously is remarkable. Through this multi-task classification, the proposed method not only maintains high performance but also proves to be more efficient than conventional approaches in handling the classification of multiple tasks concurrently.

### 4.3.3. Performance of the Efficiency

The proposed method utilizes MTL to perform multi-task classification on the ISCX VPN/Non-VPN 2016 dataset. The goal is to achieve high performance by simultaneously

handling various classification tasks. The efficiency of the model refers to its ability to quickly adapt to downstream tasks. To evaluate this efficiency, we compared the proposed method with other approaches and measured the processing speed. However, the interpretation of the model's efficiency may vary depending on hardware performance and data. Therefore, maintaining the same experimental environment and dataset is crucial for a fair comparison. Since it is difficult to reproduce these conditions exactly, we compare our results to those presented in other studies [8,33,34,39]. Table 5 shows the results on fine-tune efficiency evaluation.

**Table 5.** Results on efficiency evaluation.

| Method | Task | ST (ms) | PT (ms) |
|---|---|---|---|
| ET-BERT (Fine-Tune) [34] | STL (1 task) | 8.30~9.61 | 155.7 |
| XENTC [33] | STL (1 task) | - | 15.1 |
| MATEC [39] | STL (1 task) | 2.10 | 1.3 |
| FastTraffic [8] | STL (1 task) | 0.25 | 0.59 |
| Proposed | MTL (3 tasks) | 1.27 | 30.7 |

In Table 5, ST represents the processing time for one sample and PT represents the processing time for one packet. Among the four models, ET-BERT and XENTC are general classification models, while MATEC and FastTraffic are models designed for lightweight purposes. All of them perform single-task classification.

From an ST perspective, ET-BERT yields a range of 8.30~9.61 ms. The lightweight models, MATEC and FastTraffic, yield 2.10 and 0.25 ms, respectively. The proposed method achieves higher efficiency than ET-BERT with an execution time of 1.27 ms, but it is less efficient than MATEC and FastTraffic. Nevertheless, considering the results in Tables 3 and 4, the proposed method demonstrates 4.65~27.68% higher accuracy compared to MATEC and FastTraffic. Furthermore, the proposed method is more efficient than MATEC as it can learn the three tasks simultaneously.

From a PT perspective, ET-BERT yields 155.7 ms, XENTC produces 15.1 ms, MATEC results in 1.3 ms, and FastTraffic yields 0.59 ms. The proposed method achieves an efficiency of 30.7, which is higher than ET-BERT but lower than XENTC and FastTraffic. The proposed method seems to exhibit relatively high PT since it processes eight packets within the flow. However, similar to the ST perspective, the proposed method demonstrates high efficiency considering both accuracy and multi-task classification.

The efficiency of a model is highly influenced by hardware performance and data structure, and there is typically a trade-off between model performance and efficiency. Considering this trade-off, evaluating the balance between model performance and efficiency becomes crucial. Further discussion is needed based on additional experimental results to better understand this trade-off and assess the overall performance and efficiency of the model. Therefore, in the future, we plan to enhance the model to achieve higher efficiency while maintaining its classification performance.

## 5. Discussions

In this paper, we demonstrate high performance and efficiency by performing multi-task classification on encrypted traffic. In this section, we provide some detailed discussion of the proposed method.

### 5.1. Effect of Class Wight in Data Imbalance

We applied class weights to address the class imbalance in ISCX 2016 VPN/Non-VPN data in Section 3.2.1. Class weight represents a weight that reflects the proportion of classes in the data and can reduce the imbalance between classes. Figure 7 shows the distribution of data for each class before and after applying class weight.

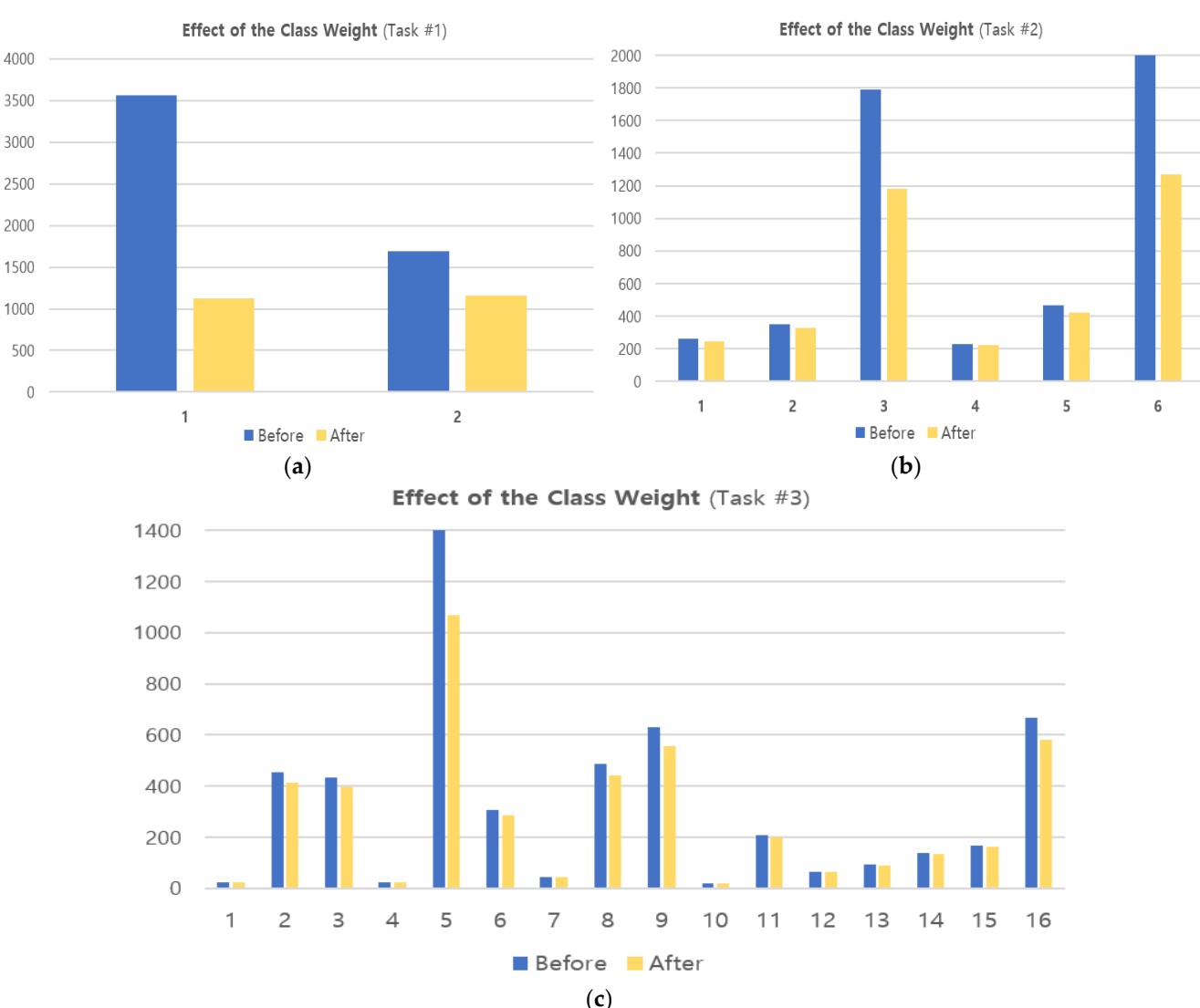

**Figure 7.** Distribution of data before and after applying class weights. (**a**) Encapsulation task (two classes), (**b**) category task (six classes), (**c**) application task (sixteen classes).

In Figure 7, the *x*-axis represents the classes per task and is organized the same as in Figure 3. For example, in Figure 7a, "1" and "2" represent {VPN, NonVPN}, respectively, and in Figure 7b, "1~6" represent {Chat, Email, File Transfer, P2P, Streaming, VoIP}. Comparing the 'Before' and 'After' in Figure 7, we can see that the imbalance between each class is significantly reduced. However, in Figure 7c, we can see that there is still some imbalance as there are too few minority classes. These limitations will be tackled in the future with additional weighting and sampling techniques.

### 5.2. Performance Based on Weight Adjustment

In order to address both data imbalance issues and variations in difficulty across tasks, we applied weight adjustments during the experiments. The weight adjustment is implemented in two aspects: class weights and task weights. Class weights were introduced to mitigate data imbalance problems, while task weights were designed to prevent biased learning, particularly when there were significant differences in difficulty among tasks. The proper utilization of these two weights is crucial, especially in scenarios where specific tasks converge rapidly; failure to handle this appropriately may lead to biased learning. Figure 8 shows the test accuracy curve for the weight adjustment.

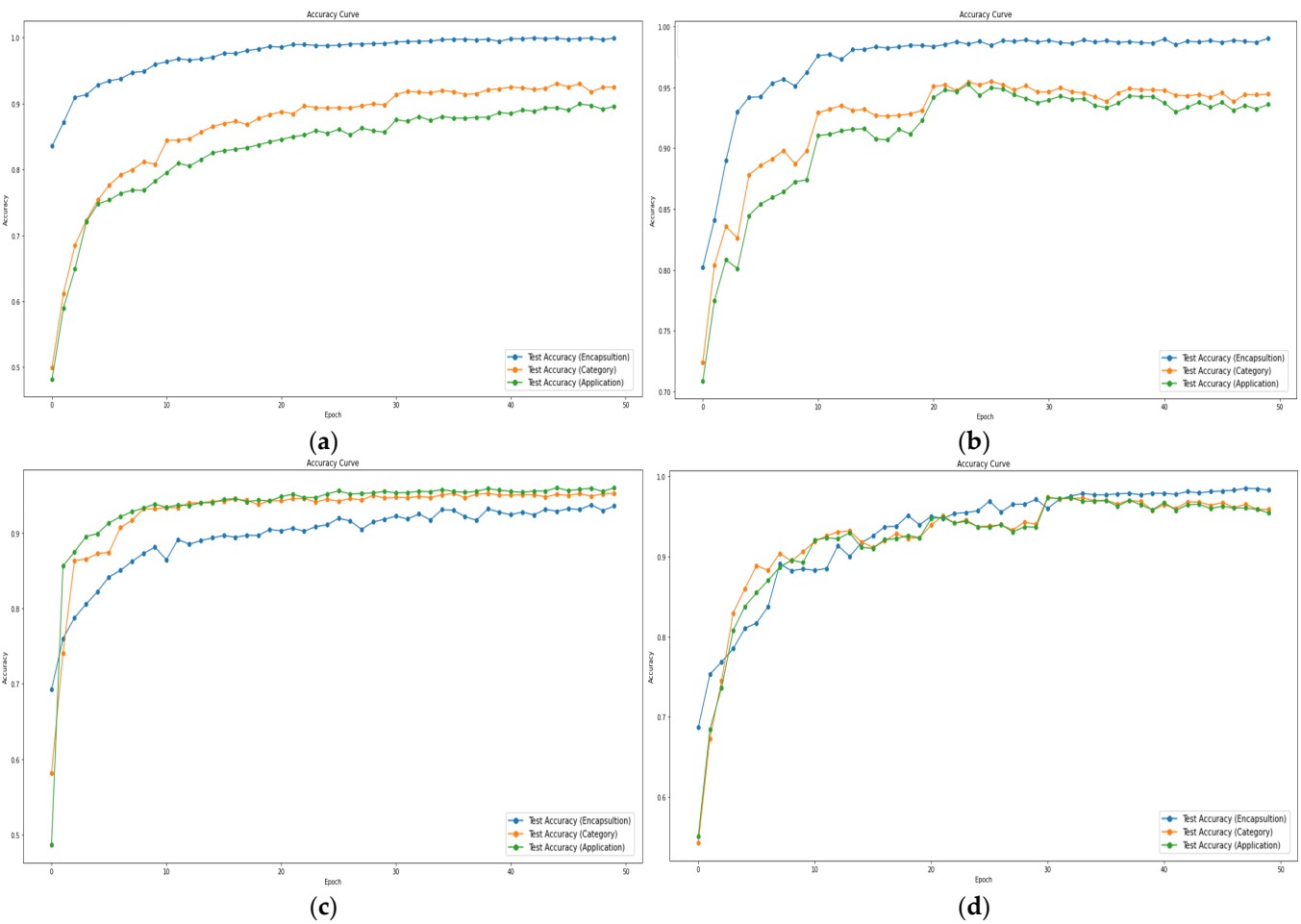

**Figure 8.** Test accuracy curve for weight adjustment. (**a**) No weight, (**b**) class weight, (**c**) task weight, and (**d**) class and task weights applied.

Figure 8a shows the results with weight adjustments, applying only class weight, yielding an accuracy of 99% across task #1, 90~91% across task #2, and 89~90% across the task #3. Figure 8b shows the results with weight adjustments, applying only class weight, yielding an accuracy of 98% across task #1 and 93–94% across tasks #2 and #3. Figure 8c shows the results with weight adjustments, applying only task weight, yielding an accuracy of 93% across task #1 and 94~95% across tasks #2 and #3. In Figure 8a,b, rapid convergence is observed in task #1, while Figure 8c tends to exhibit initially similar convergence speeds across each task, later showing higher performance in task #2 and #3 compared to task #1. Figure 8d shows the results with weight adjustments, applying class and task weights, also yielding an accuracy of 98~99% across task #1 and 96–97% across tasks #2 and #3. Through the above experiments, it is evident that adjusting weights for both categories leads to higher performance. Therefore, it can be concluded that weight adjustment plays a crucial role in enhancing performance.

### 5.3. Performance Based on Input Shape

In Section 3.1.1, we described that we conducted several experiments with various input shapes based on the number of packets and bytes in the flow. Through these experiments, we set the optimal shape as 8 packets and 63 bytes. In this section, we compare the performance based on different input shapes. The input shape can be variably defined, and we set the range of packet counts to 4~8 and byte counts to 60~70, taking into account the handshake process and header (IP, TCP/UDP) byte sizes within encrypted communication.

As mentioned earlier, considering BERT's maximum input token limit of 512, we excluded cases where the total token count (packet count × byte count) exceeds 512.

Table A1 in Appendix A indicates the performance of the proposed method based on input shapes. The experiments were conducted for 20 epochs with the same experimental setup, as multiple experiments were required depending on the input shape. We selected the target task as the most challenging task #3 among the three tasks. In Table A1, the highest performance is observed with (8, 63). Therefore, we selected the optimal input shape as 8 packets and 63 bytes.

## 6. Conclusions

Network traffic classification has been studied for a long time, and recently, a lot of research has been conducted on encrypted traffic. Most studies perform single-task classification, with DL- and transformer-based methods performing well. However, there are limitations in their efficiency and effectiveness given the increasingly diverse and complicated nature of traffic.

In this paper, we proposed multitask classification by using DistilBERT. The proposed method can learn multiple tasks in one model with one training. We applied a weight adjustment to improve the performance of our proposed method. The weight adjustment consists of class weights and task weights. Class weights mitigate the problem of data imbalance, and task weights prevent biased learning due to the difference in difficulty between tasks in multi-task classification.

To evaluate the proposed method, we conducted experiments in terms of accuracy and efficiency. Measured in terms of accuracy, the proposed approach achieves 96.89–99.29% accuracy on three tasks, showing higher performance compared to most existing methods. Furthermore, in terms of efficiency, it outperforms ET-BERT. While the proposed method exhibits lower efficiency compared to FastTraffic and MATEC, which focus on lightweight design, it achieves a significantly higher accuracy, ranging from 4.65 to 27.68% higher than the two mentioned methods. We discussed the performance impact of class weight and weight adjustment in Section 5. In addition, we validated the decision to select 8 packets and 63 bytes based on performance experiments with input data shapes (in Appendix A, Table A1). This input shape consists of packets generated during the handshake process within TLS, which is the most widely utilized today and typically remains unencrypted. Therefore, we believe it performs well despite being encrypted traffic.

However, the proposed method has some limitations. First, although the proposed method demonstrated high performance on the ISCX 2016 VPN/Non-VPN dataset, validation was only conducted on specific datasets. As the ISCX 2016 VPN/Non-VPN dataset comprises a small amount of data, leveraging AI models may yield high performance. Therefore, additional validation experiments on other datasets such as ISCX Tor are necessary to verify the performance of the proposed method. Second, as mentioned earlier, efficiency can vary depending on hardware performance and the dataset. In this paper, we evaluated the method using results presented in other studies; however, for a precise assessment, consistent experimental conditions and preprocessed datasets are necessary. Third, as previously mentioned, the proposed method utilizes eight packets in the flow, which results in a relatively high time to process a single packet. Fourth, the proposed method applies class weights to address the problem of imbalanced data. Although the class weights alleviate the problem of imbalanced data to some extent, they are still unevenly distributed. Nevertheless, the proposed method can perform three tests with one training and shows high performance and efficiency.

In future research, we plan to perform multi-task classification using diverse datasets. We will assess the effectiveness of our proposed method using identical experimental setups and preprocessed datasets for evaluation. Additionally, we plan to improve the model architecture and preprocessing methods to further enhance the performance and efficiency of the proposed method, including PT.

**Author Contributions:** Conceptualization, M.-S.K.; methodology, J.-T.P.; software, J.-T.P. and C.-Y.S.; resources, J.-T.P., C.-Y.S. and U.-J.B.; data processing, U.-J.B. and J.-T.P. writing—original draft preparation, J.-T.P.; writing—review and editing, J.-T.P.; visualization, J.-T.P.; supervision, M.-S.K.; project administration, M.-S.K.; funding acquisition, M.-S.K. All authors have read and agreed to the published version of the manuscript.

**Funding:** This work was supported by Institute of Information & communications Technology Planning & Evaluation (IITP) funded by the Korea government (00235509, Development of security monitoring technology based network behavior against encrypted cyber threats in ICT convergence environment).

**Institutional Review Board Statement:** Not applicable.

**Informed Consent Statement:** Not applicable.

**Data Availability Statement:** Data are contained within the article.

**Conflicts of Interest:** The authors declare no conflicts of interest.

## Appendix A

**Table A1.** Performance based on different input shapes.

| Performance Based on Input Shape (for Task #3: Application Classification) | | | | |
|---|---|---|---|---|
| Input Shape (Packet, Byte) | Accuracy (%) | Precision (%) | Recall (%) | F1-Score (%) |
| (4, 60) | 78.13 | 90.58 | 74.64 | 81.84 |
| (4, 61) | 85.36 | 97.14 | 83.59 | 89.85 |
| (4, 62) | 84.78 | 94.19 | 82.51 | 87.96 |
| (4, 63) | 85.10 | 91.14 | 78.57 | 84.39 |
| (4, 64) | 84.97 | 97.54 | 80.63 | 88.28 |
| (4, 65) | 85.62 | 96.52 | 81.99 | 88.66 |
| (4, 66) | 83.89 | 97.58 | 79.65 | 87.71 |
| (4, 67) | 85.32 | 93.80 | 80.48 | 86.63 |
| (4, 68) | 86.46 | 97.41 | 80.93 | 86.41 |
| (4, 69) | 86.27 | 95.07 | 81.21 | 87.60 |
| (4, 70) | 86.40 | 96.52 | 81.99 | 88.66 |
| (5, 60) | 84.24 | 96.10 | 80.34 | 87.51 |
| (5, 61) | 85.23 | 97.58 | 80.13 | 87.99 |
| (5, 62) | 83.46 | 97.57 | 80.61 | 88.28 |
| (5, 63) | 86.46 | 96.77 | 82.63 | 89.14 |
| (5, 64) | 84.93 | 97.49 | 81.82 | 88.97 |
| (5, 65) | 84.63 | 92.94 | 79.14 | 85.49 |
| (5, 66) | 84.27 | 97.50 | 79.32 | 87.48 |
| (5, 67) | 82.81 | 88.00 | 74.26 | 83.55 |
| (5, 68) | 82.24 | 90.15 | 74.78 | 81.75 |
| (5, 69) | 83.35 | 91.15 | 76.79 | 83.36 |
| (5, 70) | 84.51 | 96.37 | 81.98 | 88.59 |
| (6, 60) | 81.86 | 93.14 | 78.51 | 85.20 |
| (6, 61) | 85.89 | 94.59 | 82.44 | 88.10 |
| (6, 62) | 84.49 | 97.44 | 80.13 | 87.94 |
| (6, 63) | 88.07 | 97.76 | 83.22 | 89.91 |
| (6, 64) | 85.90 | 97.51 | 81.77 | 88.95 |
| (6, 65) | 85.78 | 97.55 | 81.52 | 88.82 |
| (6, 66) | 86.46 | 97.62 | 82.37 | 89.59 |
| (6, 67) | 86.08 | 97.51 | 83.16 | 89.77 |
| (6, 68) | 86.42 | 97.59 | 81.37 | 88.74 |
| (6, 69) | 85.93 | 97.59 | 81.50 | 86.58 |
| (6, 70) | 86.46 | 96.44 | 83.25 | 89.36 |
| (7, 60) | 81.45 | 90.61 | 80.11 | 85.03 |
| (7, 61) | 84.18 | 92.27 | 81.77 | 86.70 |
| (7, 62) | 87.74 | 97.43 | 83.99 | 90.21 |
| (7, 63) | 87.42 | 97.60 | 82.26 | 89.27 |
| (7, 64) | 82.62 | 89.23 | 78.18 | 83.34 |
| (7, 65) | 86.43 | 96.59 | 82.87 | 89.20 |
| (7, 66) | 86.73 | 97.49 | 82.60 | 89.43 |
| (7, 67) | 86.35 | 97.34 | 82.87 | 89.52 |
| (7, 68) | 84.37 | 91.88 | 78.47 | 84.65 |
| (7, 69) | 85.46 | 96.86 | 84.35 | 90.17 |
| (7, 70) | 84.79 | 94.01 | 79.11 | 85.92 |
| (8, 60) | 86.86 | 96.56 | 79.62 | 87.28 |
| (8, 61) | 86.58 | 97.59 | 81.14 | 88.61 |
| (8, 62) | 88.16 | 95.39 | 82.19 | 88.30 |
| (8, 63) | 90.28 | 98.17 | 86.28 | 91.84 |

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
