# Peer review of "Fast and Accurate Multi-Task Learning for Encrypted Network Traffic Classification"

_applsci, doi:10.3390/app14073073_

Round 1
Reviewer 1 Report
Comments and Suggestions for Authors
1. Equation 2 in the paper uses different weights to calculate the loss function for different sub-models, but does this reduce the accuracy of the sub-models with smaller weights?
2. Equation 4 in the paper is set at a fixed β, which can lead to poor applicability of the model. Is there a better way to adjust the β-value?
3. The model of this strategy is only based on the specified unique dataset, does this lead to less generalization of the model?
4. Can this method be used for classifying encrypted traffic based on different encryption methods and how does it differ from traditional traffic classification methods?
5. It is recommended that the shared layers in Figure 4 be drawn in more detail, and the role of each layer can be briefly explained in the figure.
6. The layout of some pictures and tables in the paper needs to be readjusted, such as the size and dimensions of Figure 5 and Table 4, and the position of markers in Figure 8, etc. Secondly, font formatting, paragraph numbering and text color also need to be harmonized, for example, the alignment of numbering in the contribution section on page 2, and the color of punctuation on page 8, etc.
Comments on the Quality of English LanguageModerate editing of English language required.
Author Response
We appreciate your kind and valuable review opinions. Our responses to your comments is in the attachment.
Please see the attachment

Reviewer 2 Report
Comments and Suggestions for Authors
Monitoring and regulating network traffic is an important task to ensure fast and reliable access to the required (and paid) network resources of legitimate users. Traditional methods in traffic classification have been hindered by the widespread use of encrypted payload; in those cases the classifier can rely only on the significantly less informative metadata contained in the headers of the packets. To address this issue several classification methods using tailored AI methods have been proposed and investigated in quite an abundance of research papers. To evaluate the performance and transferability of a proposed classifier poses serious problems which have not been addressed properly in this paper. The main problem is the scarcity of reliable labeled data on which such a proposed classifier can be tested. For both training and checking almost all papers use exclusively the ISCX 2016 VPN Non-VPN dataset. To recover some of the temporal characteristics of network flow, the raw packet capture data is split into so-called flows to encapsulate several packets with the same source and destination. The total number of flows in the above dataset is the sheer number 309k (page 7 of the reviewed paper). Even this small number has been cut further to a mere 8,763 (sic!) flows after filtering out flows which “were considered non-essential for the research objectives” (page 7, line 22). Any decent AI method produces a perfect fit model for such a small dataset. It takes some effort and ingenuity to scale the AI model down to fit only 99.45% (Table 2, page 13), such as masking the IP port to zero, saying “it can cause biased interpolation as it has strong identifying information” (page 8). Checking the result on real traffic data independent of the training one is imperative when training data is so sparse. My uneducated guess is that the proposed algorithm would not do significantly better than any random guess in that case.
The main methodological problem with the paper under review, however, is something completely different. Encrypted payload is guaranteed to be random, and completely unrelated to the desired classification. The best strategy is to ignore such random noise. The method proposed by the paper has a preprocessing stage (top of page 8) which extracts 63 bytes of data from each network packet, including its payload. A typical TCP header contains strictly less than 20 bytes of relevant information (actually much less as, for example, the control sum is totally irrelevant to the classification, IP addresses and port numbers are highly redundant, etc.), which means that in this stage more than 320 bits of completely random data are injected into each token.
As it has been observed by many students, machine learning always identifies patterns in completely random data. The typical reaction is that it is the result of overfitting. This is not completely true. In contrast to the general belief, random data actually exhibits patterns, albeit on a small scale. This phenomenon is well-known among psychologists and has been studied extensively. (Think of the lucky series in gambling. The general belief in no structure is witnessed by the real random Apple music playlist. For a connection to AI see Smith, Gary, 'Patterns in Randomness', The AI Delusion (Oxford, 2018; online edn, Oxford Academic, 12 Nov. 2020), https://doi.org/10.1093/oso/9780198824305.003.0007.) The injected random data is more than enough to introduce the small scale regularity (there are only 8,763 cases!) that the proposed algorithm can find and harvest. The more random data is injected, the easier is to find the desired pattern. Of course, the procedure would demonstrably fail on larger, independent test data.
As a side note, the random values used to train the model in the paper are hard-plugged into the ISCX 2016 VPN Non-VPN dataset. As these “random numbers” are fixed, and are unique, they are only expected to behave as any other choices of random numbers, but they must not necessarily do so. This particular random sequence may exhibit non-typical properties (but the probability that this happens is extremely small). An interesting research question would be to investigate how much random data suffices to convince the algorithm developed by the authors that it could do the required classification task with very high confidentiality.
Round 2
Reviewer 2 Report
Comments and Suggestions for Authors
The authors responded to all my concerns, explaining that the proposed algorithm still needs additional validation on independent data. I appreciate the Appendix clarifying the significance and usefulness of the handshake information (which is unencrypted) for the classification.